# Can Calibration Improve Sample Prioritization?

**Ganesh Tata**[*]
University of Alberta
`gtata@ualberta.ca`

**Gautham Krishna Gudur**[*]
Global AI Accelerator, Ericsson
`gautham.krishna.gudur@ericsson.com`

**Gopinath Chennupati**
Amazon Alexa
`cgnath.dr@gmail.com`

**Mohammad Emtiyaz Khan**
RIKEN Center for AI Project
`emtiyaz.khan@riken.jp`

## Abstract

Calibration can reduce overconfident predictions of deep neural networks, but *can calibration also accelerate training?* In this paper, we show that it can when used to prioritize some examples for performing subset selection. We study the effect of popular calibration techniques in selecting better subsets of samples during training (also called sample prioritization) and observe that calibration can improve the quality of subsets, reduce the number of examples per epoch (by at least 70%), and can thereby speed up the overall training process. We further study the effect of using calibrated pre-trained models coupled with calibration during training to guide sample prioritization, which again seems to improve the quality of samples selected.

## 1    Introduction

Calibration is a widely used technique in machine learning to reduce overconfidence in predictions. Modern deep neural networks are known to be overconfident classifiers or predictors, and calibrated networks provide trustworthy and reliable confidence estimates [1]. Hence, finding new calibration techniques and improving them has been an active area of research [1, 6, 10, 11].

In this paper, we ask *if calibration aids in accelerating training by using sample prioritization*, i.e., we select training samples based on calibrated predictions to better steer the training performance. We explore different calibration techniques and focus on selecting a subset with the most informative samples during each epoch. We observe that *calibration performed during training plays a crucial role in choosing the most informative subsets*, which in turn accelerates neural network training. We then investigate the effect of an external pre-trained model which is well-calibrated (with larger capacity) on the sample selection process during training.

Our contributions are as follows,

We provide an in-depth study analyzing the effect of various calibration techniques on sample prioritization during training. We also consider pre-trained calibrated *target* models and observe their effect on sample prioritization along with calibration during training. We benchmark our findings on widely used CIFAR-10 and CIFAR-100 datasets and observe the improved quality of the chosen subsets across different subset sizes, which ensures faster deep neural network training.

---

[*]Both authors contributed equally to this work.

Has it Trained Yet? Workshop at the Conference on Neural Information Processing Systems (NeurIPS 2022).

## 2 Background

### 2.1 Problem Statement

We formulate the problem in the paper as follows. A calibration technique $C$ is performed during training at each epoch, and a sample prioritization function $a$ is then used to select the most informative samples for training each subsequent epoch. We use *Expected Calibration Error (ECE)* for model calibration [10], which measures the absolute difference between the model's accuracy and its confidence.

The paper discusses how a calibration technique $C$, when coupled with a sample prioritization function $a$, affects the performance (accuracy and calibration error (ECE)) of the model. In addition, we also observe if this phenomenon can aid in faster and more efficient training. We hypothesize a closer relationship between calibration and sample prioritization during training, wherein the calibrated model probabilities at each epoch are used by a sample prioritization criterion to select the most informative samples for training each subsequent epoch.

### 2.2 Calibration

Calibration is a technique that curbs overconfident predictions in deep neural networks, wherein the predicted (softmax) probabilities reflect true probabilities of correctness (better confidence estimates) [1]. In this paper, we consider various prominently used calibration techniques which are performed during training.

***Label Smoothing*** implicitly calibrates a model by discouraging overconfident prediction probabilities during training [9]. The one-hot encoded ground truth labels ($y_k$) are smoothened using a parameter $\alpha$, that is $y_k^{LS} = y_k(1 - \alpha) + \alpha/K$, where $K$ is the number of classes. These smoothened targets $y_k^{LS}$ and predicted outputs $p_k$ are then used to minimize the cross-entropy loss.

***Mixup*** is a data augmentation method [14] which is shown to output well-calibrated predictive scores [13], and is again performed during training.

$$\bar{x} = \lambda x_i + (1 - \lambda)x_j$$
$$\bar{y} = \lambda y_i + (1 - \lambda)y_j$$

where $x_i$ and $x_j$ are two input data points that are randomly sampled, and $y_i$ and $y_j$ are their respective one-hot encoded labels. Here, $\lambda \sim \text{Beta}(\alpha, \alpha)$ with $\lambda \in [0, 1]$.

***Focal Loss*** is an alternative loss function to cross-entropy which yields calibrated probabilities by minimizing a regularized KL divergence between the predicted and target distributions [8].

$$L_{Focal} = -(1 - p)^{\gamma} log p$$

where $p$ is the probability assigned by the model to the ground-truth correct class, and $\gamma$ is a hyperparameter. When compared with cross-entropy, Focal Loss has an added factor that encourages the samples predicted with correct classes to have lower probabilities. This enables the predicted distribution to have higher entropy, thereby helping avoid overconfident predictions.

### 2.3 Sample Prioritization

Sample prioritization is the process of selecting important samples during different stages of training to accelerate the training process of a deep neural network without compromising on performance. In this paper, we perform sample prioritization during training using *Max Entropy*, which is a de facto uncertainty sampling technique to select the most efficient samples at each epoch.

***Max Entropy*** selects the most informative samples (top-$k$) that maximize the predictive entropy [12].

$$\mathbb{H}[y|x, D_{train}] := -\sum_c p(y = c|x, D_{train}) \log p(y = c|x, D_{train})$$

### 2.4 Pre-trained Calibrated Target models

Pre-trained models have been widely used in literature to obtain comprehensive sample representations before training a downstream task [5]. We use a *pre-trained calibrated model with larger capacity*

which we call the *target* model [3] and use the Max Entropy estimates obtained from this target model at each epoch to select the samples, thereby guiding the corresponding epochs of the model during training. Further, we call the model which is being trained as the *current* model. In this paper, we perform sample prioritization with the target model in addition to calibrating the current model.

Table 1: Test Accuracies (%) and ECEs (%) across various calibration techniques and subset sizes with Resnet-34 as *current* model for both datasets.

| Dataset | Calibration | 100% | | 30% | | 20% | | 10% | |
|---|---|---|---|---|---|---|---|---|---|
| | | Accuracy | ECE | Accuracy | ECE | Accuracy | ECE | Accuracy | ECE |
| CIFAR-10 | **No Calibration** Cross-Entropy (Baseline) | 94.1 | 4.1 | 93.6 | 5.33 | **93.86** | 4.01 | **93.23** | 5.2 |
| | **Label Smoothing** 0.03/0.05/0.05/0.03 | 94 | 1.84 | 91.74 | 3.17 | 91.48 | 3.56 | 91.72 | 2.71 |
| | **Mixup** 0.1/0.3/0.2/0.15 | **95.1** | 2.1 | **94.39** | 2.67 | 93.35 | 2.59 | 93.17 | 1.78 |
| | **Focal Loss** 1/3/3/3 | 94.69 | **1.71** | 93.19 | **1.2** | 92.6 | **1.25** | 92.25 | **1.42** |
| CIFAR-100 | **No Calibration** Cross-Entropy (Baseline) | 77.48 | 5.42 | 73.13 | 10.77 | 71.54 | 13.16 | **69.65** | 14.47 |
| | **Label Smoothing** 0.03/0.03/0.03/0.09 | 77.05 | 4.88 | 72.21 | 3.45 | 70.93 | 5.75 | 68.63 | 5.67 |
| | **Mixup** 0.15/0.15/0.15/0.35 | **78.68** | 3.59 | **73.57** | **1.49** | **72.02** | **2.4** | 69.1 | **1.16** |
| | **Focal Loss** 1/3/3/5 | 78.59 | **3.57** | 71.86 | 1.67 | 70.61 | 3.25 | 65.81 | 1.82 |

## 3 Experiments and Results

We perform our experiments on CIFAR-10 and CIFAR-100 [4] datasets with the setting mentioned in Section 2.1, and we use Resnet-34 [2] as the *current* model. For both datasets, the initial training set consisting of 50,000 samples is split into 90% training data and 10% validation data, while the test set contains 10,000 samples. In the sample prioritization setting, we start with 10 warm-up epochs in which all samples are selected during training (no subset selection), following which we select $n\%$ of total training samples in each epoch using the Max Entropy criterion. We choose different settings of subset sizes for each epoch with $n - \{10, 20, 30\}$.

We use the Stochastic Gradient Descent optimizer with initial learning rates of 0.01 and 0.1 for CIFAR-10 and CIFAR-100 respectively, trained for 200 epochs with a cosine annealing [7] scheduler, weight decay of $5e^{-4}$ and momentum of $0.9$ for both datasets. The models are trained using a V100 GPU. We consider classification accuracies and ECEs across different calibration techniques with their respective parameter sweeps as follows: *Label Smoothing* ($\alpha$) – {0.01, 0.03, 0.05, 0.07, 0.09}, *Mixup* ($\alpha$) – {0.1, 0.15, 0.2, 0.25, 0.3, 0.35} and *Focal Loss* ($\gamma$) – {1, 2, 3, 4, 5}. As baselines, we consider uncalibrated models with standard cross-entropy loss. For the *target* experiments, we choose Resnet-50 models [2] trained with Mixup (with $\alpha = 0.3$ for CIFAR-10, and $\alpha = 0.25$ for CIFAR-100) as our target models after performing parameter sweeps across all calibration techniques.

### 3.1 Discussion on Results

Table 1 shows the test ECEs and accuracies for the current model across different calibration techniques and subset sizes with Max Entropy criterion. We can observe that *all calibration techniques have lower test ECEs than their respective uncalibrated models across all subset sizes* for both datasets. This demonstrates that *performing calibration during training improves sample prioritization*. Moreover, these results indicate that there are no significant trade-offs between model accuracy and model confidence (ECE) when calibration is performed with sample prioritization. Figures 1a and 1b also illustrate lower validation ECEs across training epochs for calibrated current models when compared to their uncalibrated counterparts, with 30% subset size as an instance. In particular, we observe that test accuracies for Mixup across all subset sizes are consistently comparable and often higher than their respective uncalibrated models.

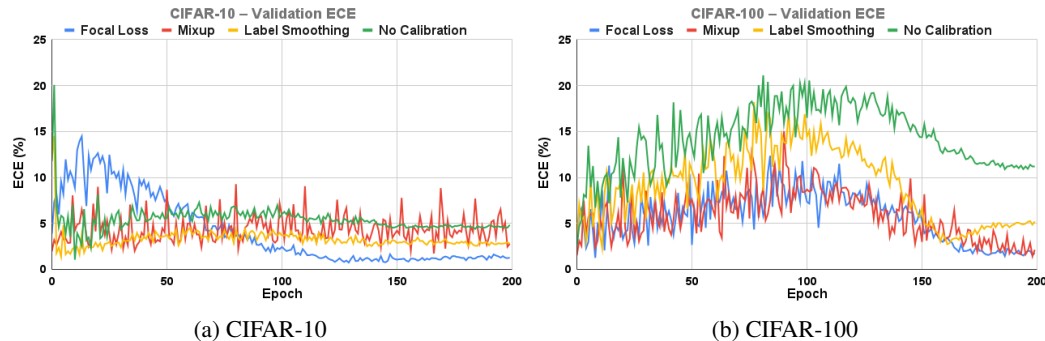

(a) CIFAR-10             (b) CIFAR-100

Figure 1: Validation ECEs (%) for both datasets with 30% subset size for *current* model.

Table 2: Test Accuracies (%) and ECEs (%) across various calibration techniques and subset sizes with Resnet-34 as *current* model for both datasets, and Resnet-50 (Mixup) as *target* model.

| Dataset | Calibration | 100% | | 30% | | 20% | | 10% | |
|---|---|---|---|---|---|---|---|---|---|
| | | Accuracy | ECE | Accuracy | ECE | Accuracy | ECE | Accuracy | ECE |
| CIFAR-10 | **No Calibration** 
 Cross-Entropy (Baseline) | 94.1 | 4.1 | 93.95 | 4.04 | 93.43 | 4.9 | 93.16 | 4.11 |
| | **Label Smoothing** 
 0.03/0.05/0.05/0.03 | 94 | 1.84 | 93.62 | 2.93 | 93.3 | 3.32 | **93.27** | 1.9 |
| | **Mixup** 
 0.1/0.3/0.15/0.15 | **95.1** | 2.1 | **94.7** | 2.88 | **93.79** | 2.73 | 93.22 | 2.16 |
| | **Focal Loss** 
 1/2/2/1 | 94.69 | **1.71** | 93.15 | **1.06** | 92.65 | **1.58** | 92.84 | **1.89** |
| CIFAR-100 | **No Calibration** 
 Cross-Entropy (Baseline) | 77.48 | 5.42 | 75.38 | 9.36 | 75.04 | 9.39 | 71.07 | 9.27 |
| | **Label Smoothing** 
 0.03/0.03/0.03/0.09 | 77.05 | 4.88 | **76.06** | 2.28 | **75.27** | 2.67 | **72.59** | 1.63 |
| | **Mixup** 
 0.15/0.2/0.15/0.15 | **78.68** | 3.59 | 75.62 | **0.86** | 74.78 | **1.43** | 70.32 | **0.86** |
| | **Focal Loss** 
 1/2/3/2 | 78.59 | **3.57** | 74.89 | 2.37 | 73.73 | **1.43** | 70.89 | 1.51 |

As expected, we observe from Table 1 that the test accuracies reduce when the subset size becomes smaller. Moreover, Mixup consistently has higher test accuracies and low test ECEs across different subset sizes compared to other calibration techniques for both datasets. This could be attributed to Mixup being a data transformation/augmentation technique, thereby performing well even in the low-data regime. Further, Figure 2 shows that training with Mixup leads to a relatively balanced representation of classes in the chosen subsets. In contrast, Label Smoothing and Focal Loss are loss-based calibration techniques with no explicit transformation performed on the underlying training data. Here, we note that Focal Loss can be more effective when coupled with a post hoc calibration technique like temperature scaling [8]. However, this setting is not applicable in our paper since we explicitly focus on calibration techniques during training. In addition, Mixup and Focal Loss are known to outperform Label Smoothing [13, 8].

***Effectiveness of Target:*** Table 2 exhibits the results when a pre-trained calibrated target model (Resnet-50 with Mixup) is used for guiding the current model while performing sample prioritization with calibration during training. Interestingly, a well-calibrated target model can boost the performance on both datasets for under-performing calibration techniques (like Label Smoothing) performed only on the current model without the target's influence. However, for calibration techniques that are already performing well (like Mixup and Focal Loss), there is no significant loss in performance on CIFAR-10, while there is a considerable improvement in performance in general on CIFAR-100. This can be clearly observed when comparing Table 2 with Table 1. We assume that the performance of a current model trained with 100% data is similar for all experiments performed with and without a target model for each respective dataset.

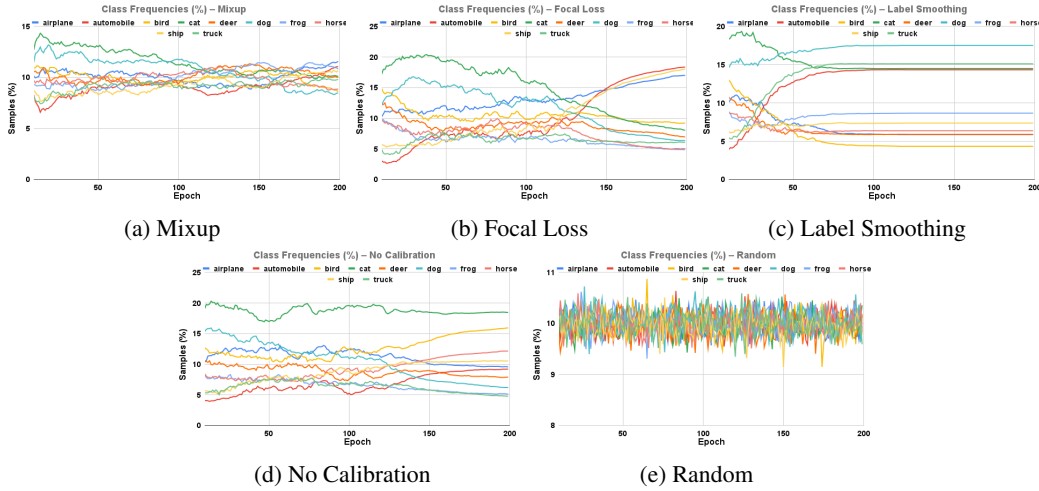

(a) Mixup      (b) Focal Loss      (c) Label Smoothing

(d) No Calibration      (e) Random

Figure 2: Class Distribution across epochs for all calibration techniques with Max Entropy (a)-(d) and Random Sampling (e) with 30% subset size for *current* model on CIFAR-10.

***Class Distribution:*** Figures 2a - 2d show the class distribution across training epochs for different calibration techniques with sample prioritization performed using Max Entropy. For comparison, we also consider Random sampling as a baseline (Figure 2e). As discussed above, performing Mixup (Figure 2a) during training leads to a relatively balanced representation of classes in each epoch. In contrast, Label Smoothing and Focal Loss (Figures 2b and 2c) do not exhibit a balanced representation of classes in the chosen subsets. The class representations are imbalanced for the uncalibrated setting as well (Figure 2d).

## 4   Conclusion

In this paper, we investigate whether existing calibration techniques improve sample prioritization during training. We empirically show that a deep neural network calibrated during training selects better subsets of samples than an uncalibrated model. We also demonstrate the effectiveness of pre-trained calibrated target models in guiding sample prioritization during training, thereby boosting calibration performance. Finally, since calibration aids in sample prioritization by improving the quality of subsets, it ensures faster neural network training.

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

# A    Common Samples between Epochs

Figures 3a and 3b show the percentage of common samples between consecutive epochs for CIFAR-10 and CIFAR-100 respectively across various calibration techniques, with Max Entropy and Random Sampling as the sample prioritization criteria with 30% subset size. In all settings, sample selection with Max Entropy leads to a significantly higher percentage of common samples between consecutive epochs throughout training, as opposed to random sampling which results in very few common samples. Although only a small percentage of samples change across epochs, sample prioritization with Max Entropy coupled with any calibration technique yields good classification performance.

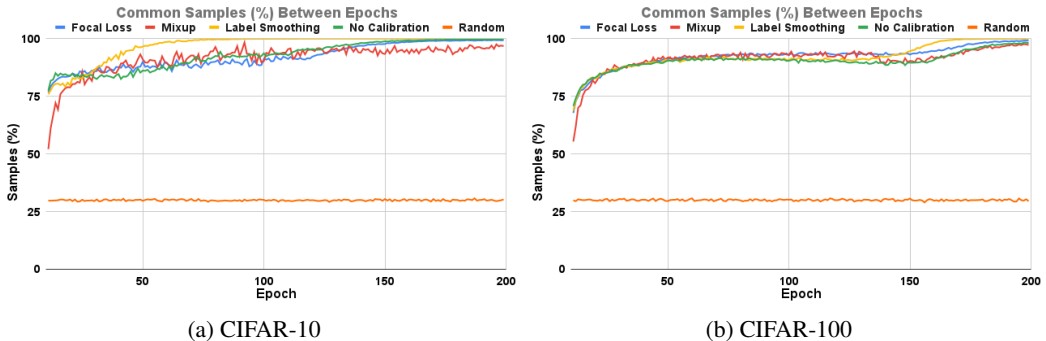

(a) CIFAR-10                                        (b) CIFAR-100

Figure 3: Common Samples (%) between epochs with 30% subset size for *current* model.

# B    Validation Accuracies

We report the validation accuracies of CIFAR-10 and CIFAR-100 with 30% subset size for the current model across all calibration techniques with Max Entropy as the sample prioritization criterion in Figures 4a and 4b.

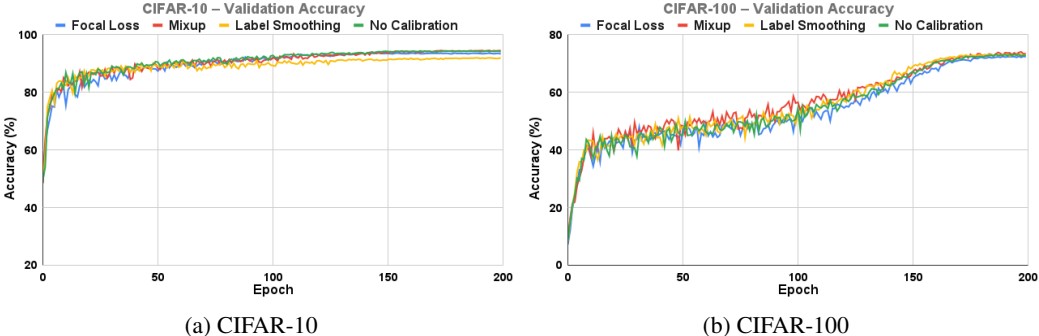

(a) CIFAR-10                                        (b) CIFAR-100

Figure 4: Validation Accuracies (%) for both datasets with 30% subset size for *current* model.

# C Reliability Diagrams

We report the reliability diagrams of CIFAR-10 and CIFAR-100 with 30% subset size for the current model across all calibration techniques with Max Entropy as the sample prioritization criterion in Figure 5 and 6.

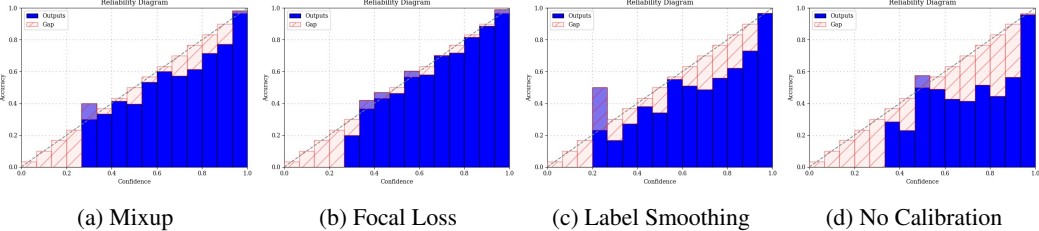

|(a) Mixup|(b) Focal Loss|(c) Label Smoothing|(d) No Calibration|

Figure 5: Reliability diagrams for CIFAR-10 with 30% subset size for *current* model.

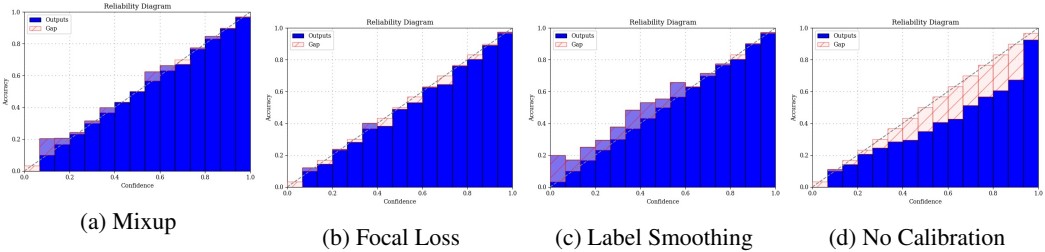

|(a) Mixup|(b) Focal Loss|(c) Label Smoothing|(d) No Calibration|

Figure 6: Reliability diagrams for CIFAR-100 with 30% subset size for *current* model.

