# OpenReview forum: "Can Calibration Improve Sample Prioritization?"
_NeurIPS.cc/2022/Workshop/HITY — HITY Workshop NeurIPS 2022_

### Official Review · Reviewer_9s7d · 2022-10-11

**Rating:** 1
**Confidence:** 4

**Review:**

This paper empirically studies whether a well-calibrated, pre-trained network can be used for sample selection during the training of another network. The idea is very intuitive: well-calibrated predictions $\implies$ good entropy estimates $\implies$ reliable selection of most informative training samples. I, therefore, think that this paper can facilitate interesting discussions.

---

### Official Review · Reviewer_kHZn · 2022-10-19

**Rating:** 1
**Confidence:** 4

**Review:**

Nice paper discussing and testing how calibration affects the performance of data prioritization.

---

### Decision · Program_Chairs · 2022-10-20

Accept